# Improvement of the Chimney Effect in Stack Ventilation

**Romana Antczak-Jarząbska** [1,*], **Krzysztof Pawłowski** [2] and **Maciej Niedostatkiewicz** [1]

[1]  Faculty of Civil and Environmental Engineering, Gdansk University of Technology, 11/12 Gabriela Narutowicza Street, 80-233 Gdańsk, Poland; mniedost@pg.edu.pl
[2]  Faculty of Civil and Environmental Engineering and Architecture, Bydgoszcz University of Science and Technology, Al. Prof. S. Kaliskiego 7, 85-796 Bydgoszcz, Poland; krzypaw@utp.edu.pl
*   Correspondence: romana.antczak@gmail.com

**Abstract:** The article is focused on the airflow in a ventilation system in a building. The work examines the methods which enhance the chimney effect. In this paper, three cases with different chimneys were analyzed for the full-scale experiment. These cases were characterized by different geometrical and material parameters, leading to differences in the intensity of the ventilation airflow. The common denominator of the cases was the room with the air inlet and outlet to the ventilation system. The differences between the experimental cases concerned the chimney canal itself, and more precisely its part protruding above the roof slope. The first experimental case concerned a ventilation canal made in a traditional way, from solid ceramic brick. The second experimental case concerned the part that led out above the roof slope with a transparent barrier, called a solar chimney. In the third experimental case, a rotary type of chimney cap was installed on the chimney to improve the efficiency of stack ventilation. All these cases were used to determine the performance of natural ventilation—Air Change per Hour (CH). Additionally, the paper presents a technical and economic comparison of the solutions used.

**Keywords:** stack ventilation; ACH; energy efficiency; solar chimney; chimney cap

## 1. Introduction

Natural ventilation (NV) is the cheapest strategy for distributing fresh air inside a building. It is still the most popular system found in Poland and other Central and Eastern European countries. Natural ventilation is driven by two physical phenomena, wind [1] and buoyancy (stack effect) caused by the temperature difference between indoor and outdoor air temperatures [2]. However, Gładyszewska et al. [3] showed that the NV is mainly affected by wind direction and velocity. The most popular type of natural ventilation used in countries with a cold climate is gravity ventilation, also known as stack ventilation. This type of natural ventilation is analyzed in this paper.

Stack ventilation should work when one or all the natural forces are available. Unfortunately, the two main drivers causing stack ventilation flows are stochastic; therefore, stack ventilation may be difficult to control and predict, as well as analyze and design. Other disadvantages of stack ventilation are the reduced control of air distribution within the building and the lack of its effectiveness in summer conditions with minimal wind. Despite the difficulty in control, stack ventilation is still relied upon to meet the need for fresh air in many buildings. The study [4] shows that stack ventilation has become a new trend in building design in the architectural community. Furthermore, stack ventilation has been used in many types of buildings, even in highly controlled indoor climate hospitals [5].

According to the standard [6], stack ventilation may cease to function properly if the outside air temperature $T_e$ exceeds 12 °C or when we talk about the so-called wind silence or the so-called light wind, which has been checked on the basis of previous own research by Antczak-Jarząbska R., Niedostatkiewicz M. [7]. To find out the number of days with unfavorable climatic conditions, the number of hours in a calendar year was checked based

on a typical Meteorological Year, during which it is highly probable that gravity ventilation will not work. The following assumptions were made for the analysis:

- Outdoor air temperature of 12 °C $\leq$ T$_e$ $\leq$ 20 °C—assumed value of 20 °C as the upper criterion, because above this temperature it was assumed that there are warm days and it is possible to ventilate the rooms.
- Wind velocity V$_{wind}$ = 0–5 m/s.

Data from a Typical Meteorological Year for the city of Elbląg was used for the analysis. Elbląg, due to its location, is not exposed to frequent gusts of wind, as are the cities located on the coast. For the city of Elbląg, continuous measurements of the external climate for 30 years were made. The Typical Meteorological Year [8] contains averaged hourly data compiled according to the ISO methodology [9]. In Figure 1, the external temperature range, during which gravity ventilation ceases to be efficient, is marked in black dots. On the other hand, the red dots show the wind velocity, which may be too low to maintain the efficiency of gravity ventilation at the desired level. The analysis shows (Figure 1) that in a year there are over 5500 h during which at least one mechanism occursand the movement of the ventilation air does not work. This is as much as 60% of the entire calendar year. On the other hand, if the mechanisms causing the movement of the ventilation air do not work at all (according to assumptions: 1. Outside air temperature 12 °C $\leq$ T$_e$ $\leq$ 20 °C and 2. Wind velocity V$_{wind}$ = 0–5 m/s), we obtain 2200 h of occurrence problems with gravity ventilation, which accounts for about 26% of the entire calendar year.

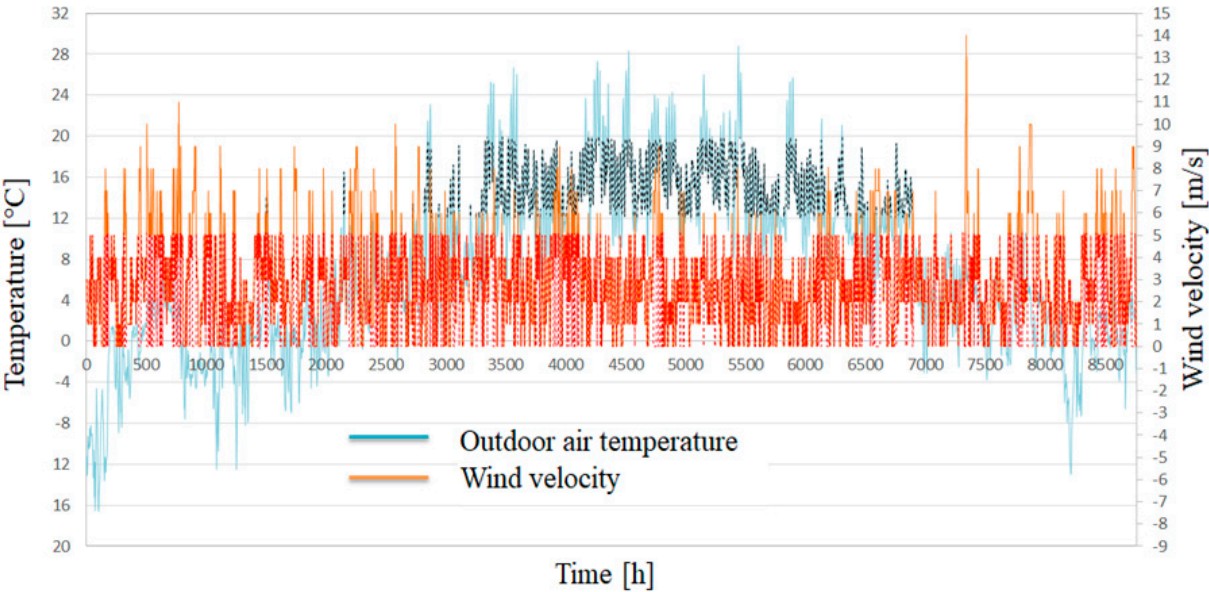

**Figure 1.** Outside air temperature and wind velocity determined for the city of Elbląg based on data from TMR.

The quality of stack ventilation is usually determined by the performance indicator of the ventilation system—air change rate (ACH). The ACH measures how quickly the air in the interior space is beingreplaced by air coming from the outside [10]. Ventilation performance indicator's roleis to provide information concerning indoor air parameters in a room or a building.

$$\text{ACH (t)} = \frac{\dot{V}(t) * 3600}{V_R} \tag{1}$$

where:

**ACH(t)**—stack ventilation times (ventilation efficiency), h$^{-1}$;
**V̇(t)**—the amount of air leaving the ventilation system, m$^3$/s;
**V$_R$**—cubature of a ventilated room, m$^3$.

The process of air exchange during the measurement was adopted as a value independent of time. Therefore, the amount of air leaving the ventilation system **V(t)** and the stack ventilation times **ACH(t)** are assumed as constant values V and ACH. The following simplifications have been introduced to determine the stack ventilation multiplicity, which is to show the ventilation efficiency:

- The air during measurements is thoroughly mixed;
- There are no gradients inside the room, meaning the air concentration is the same in the entire ventilation system.

Information about realistic climate data of the local climate conditions is important, but usually not provided. In general terms, there are two different approaches to describe the climate data: Local Climate Station (LCS) and Typical Meteorological Year (TMY). In engineering, it is uncommon to consider the Local Climate Station (LCS). As we can see in the study [11], the most common practice is to use climate data from the TMY. However, [11] while based on climate data from the TMY, such values are usually slightly overestimated.

In this paper, to determine the influence of passive energy on the stack ventilation system, in situ tests were carried out for a typical residential building located in Gdańsk. The research was conducted in the summer and autumn periods from August to November. A single-family building was used for the research. The core of the research was to experimentally check the ACH value for the solution to improve the chimney effect in stack ventilation. In this case, two different ways to reinforce the chimney effect were used: the solar chimney and chimney cap. Therefore, a room with an air inlet and an outlet to the ventilation system was separated from the building. These limitations made it possible to obtain results mainly dependent on the external climate. During the experimental research, external climate research was carried out as well, based on its own climatic station in the vicinity of the building.

The research was divided into three types. TYPE I consisted ofa test for the ventilation airflow in a traditional ventilation system, i.e., a room + ventilation duct made of solid ceramic brick. TYPE II consisted of enclosing a part of the chimney led out above the roof surface with a transparent wall. However, in TYPE III, a rotary-type chimney cap was used, which was installed on the traditional chimney described in TYPE I. During the research, the location of sensors was constant for the sensors inside the system and for the climate base.

## 2. Test House

The test building wasa two-store, two-apartment residential house located in Gdansk (northern Poland), a region of cold climate. The apartment, with a floor area of approximately 50 m², was selected for the measurement campaign (Figure 2a). The apartment was located on the first floor and consisted of a bathroom, bedroom, kitchen, living room and corridor. To simplify the ventilation system configuration, the bathroom, bedroomand corridor were excluded from the measurement space and isolated from the rest of the apartment (Figure 2b). The indoor test space was limited to the kitchen. The test apartment was inhabited during the measurement campaign. The residents' activities were registered mainly in the morning and in the evening (after 5:00 p.m.). The test house was a heavyweight building. The floor slabs were made of concrete. Stone materials are used for both sides of the cavity walls. The building was erected in the 1950s and thermally renovated in 2012. Table 1 provides the U-values overview of the main structural components.

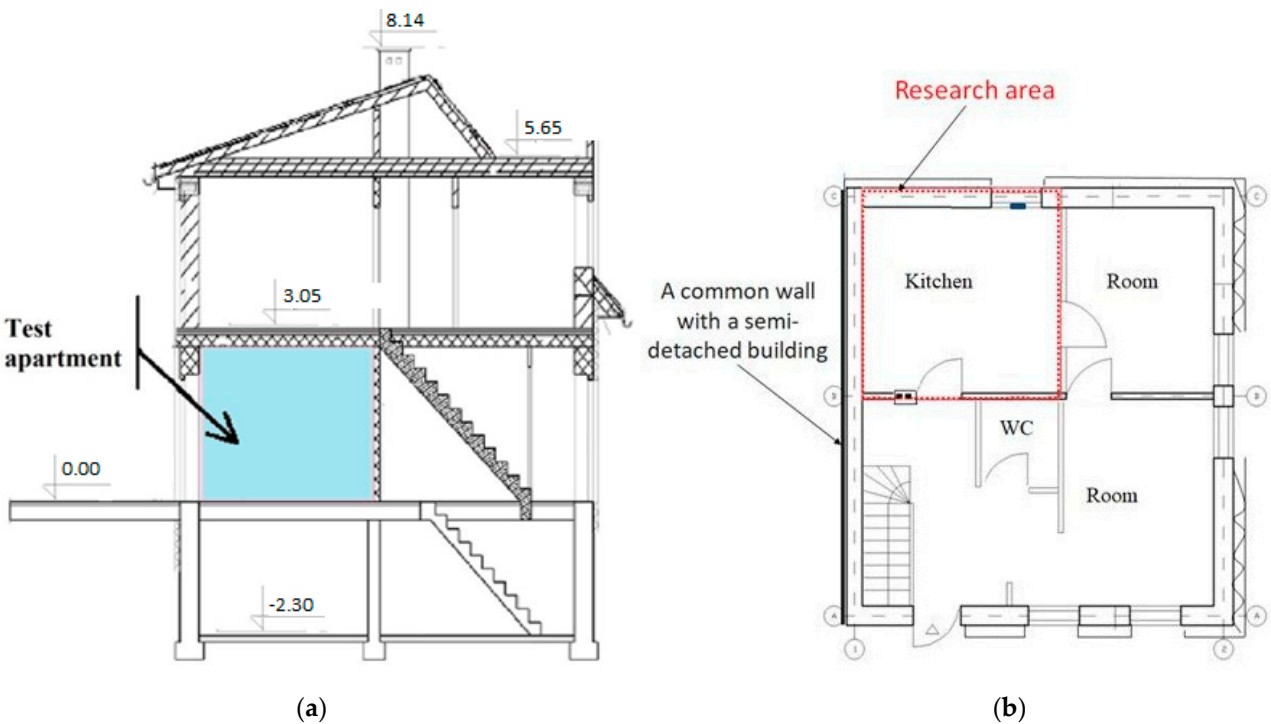

(**a**)

(**b**)

**Figure 2.** The residential family test house: (**a**) vertical section and (**b**) cross-section of the first floor.

**Table 1.** Overview of construction details for the base case building from 2012.

| Element | Details | U-Value |
|---|---|---|
| External wall | Cavity wall with (inside to outside): limestone inner layer, thermal insulation, brick outer layer | $0.18 \text{ W}/(\text{m}^2\text{K})$ |
| Door | Wood | $1.10 \text{ W}/(\text{m}^2\text{K})$ |
| Windows | Double panel glazing | $0.90 \text{ W}/(\text{m}^2\text{K})$ |

The test apartment was equipped with astack ventilation system in configuration with air inlets and chimneys ducts. Air inlets are small appliances mounted in the casement or window frame, which enablecontrol of the fresh air inflow to the room. They were invented and introduced in the 1960s in Scandinavian countries [12]. In Poland, this type of inlet gap (Figure 3a) is obligatory in the case of usage of stack ventilation in configuration with multi-chimneys ducts. The air inlet into the chimney duct was equipped with a controllable vent grill, which was located 0.15 m below the room ceiling. The measurement system was limited to the kitchen (Figure 2b) with a usable area of 15.75 m² and an air volume of 40.95 m³. The configuration with one air inlet (rectangular inlet gap in a window frame) and one air outlet (a chimney duct in the kitchen) allowed forthe reduction of the unknown parameters influencing the air exchange process. The used air was removed through a vent with dimensions of 0.14 × 0.14 m (Figure 3b). During the measurement timeperiod, all windows were closed. The doors inside the test apartment were closed during the nighttime and during the residents' worktime.

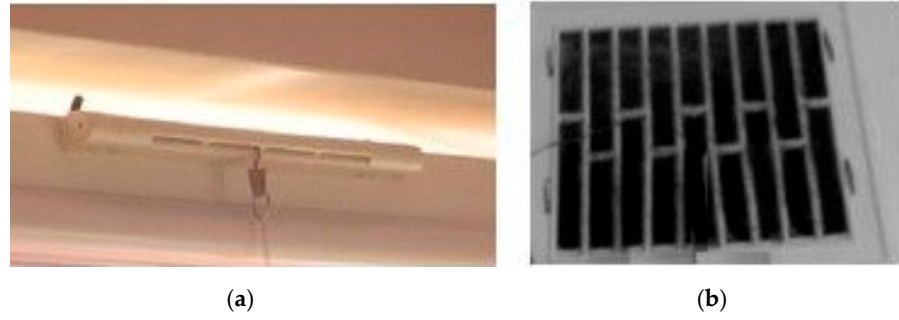

**Figure 3.** Inlet gap (**a**) and outlet vent (**b**) in research.

In the test room, there was a single-chamber PVC window with dimensions: 4/16/4 [mm]. The room had two doors, each leading to a heated room. While the residents were not present, the door was tightly closed. The chimney was made withtraditional technology, with full ceramic brick. This is a description of TYPE I (Figure 4). The chimney protruded above the roof slope to a height of 1.04 m. The building wall was made of solid ceramic bricks with a total thickness of 0.38 m. Partition walls weremade of solid ceramic bricks, 0.12 m thick.

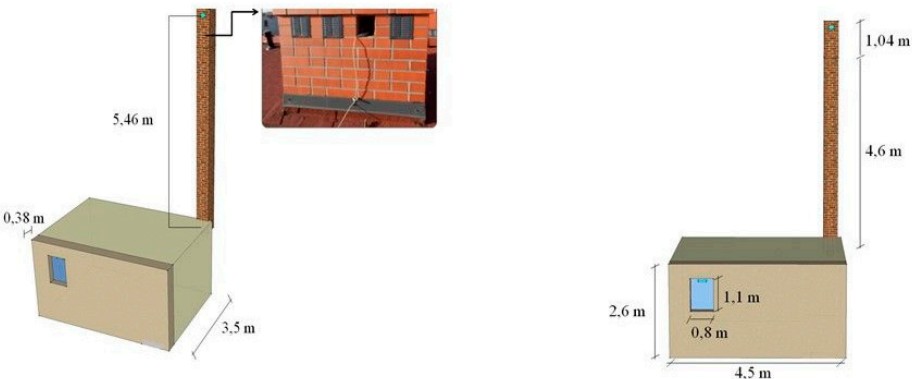

**Figure 4.** Test room and stack ventilation system, TYPE I.

In TYPE II, parts of the chimney protruding above the roof surface were enclosed with a transparent wall (float glass) with an air gap thickness of 0.04 m (Figure 5a). Glass with a thickness of 5 mm was used for the housing. In TYPE II, the remaining elements of the ventilation system did not change. The material and thermal parameters of the partitions in the ventilation system are presented in Table 2.

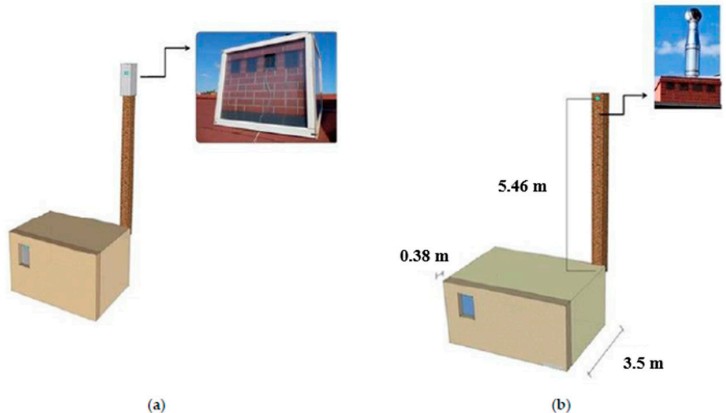

**Figure 5.** Test room and stack ventilation system, (**a**) TYPE II, (**b**) TYPE III.

**Table 2.** The material and thermal parameters of the partitions in the ventilation system.

| Element | Thickness d [m] | Thermal Conductivity λ [W/mK] [12] | Emissivity e [—] |
|---|---|---|---|
| Window joinery | 0.024 | 1.4 | 0.95 |
| External wall | 0.38 | 0.77 | 0.84 |
| Chimney wall | 0.12 | 0.77 | 0.84 |
| Glass wall | 0.005 | 6 | 0.95 |
| Air gap | 0.04 | 0.025 | - |

In TYPE III, a rotary-type chimney cap was used, which was mounted on the traditional chimney described in TYPE I (Figure 5b). The chimney cap was the second way to improve gravity ventilation in an experimental study.

## 3. Measurement Setup and Equipment

The measurement system was designed for the constant monitoring of indoor climate conditions, ambient climate conditions and airflow velocity in vent inlets and outlets. It included sensors, port data acquisition modules, a network switch and a PC computer (Figure 6). All sensors were wired directly to the data acquisition modules which registered the measured data, which in turn were connected to the PC computer via a 16-port Ethernet Switch. The measurement data management software LBX 2012 of LAB-EL [13] was installed on aPC computer. The data collected in the module data acquisition was automatically transferred to the SQL database located on a PC computer. The sensors and the data acquisition modules worked in a real-time mode, while the data management software was synchronized with the data acquisition module at each probe time step.

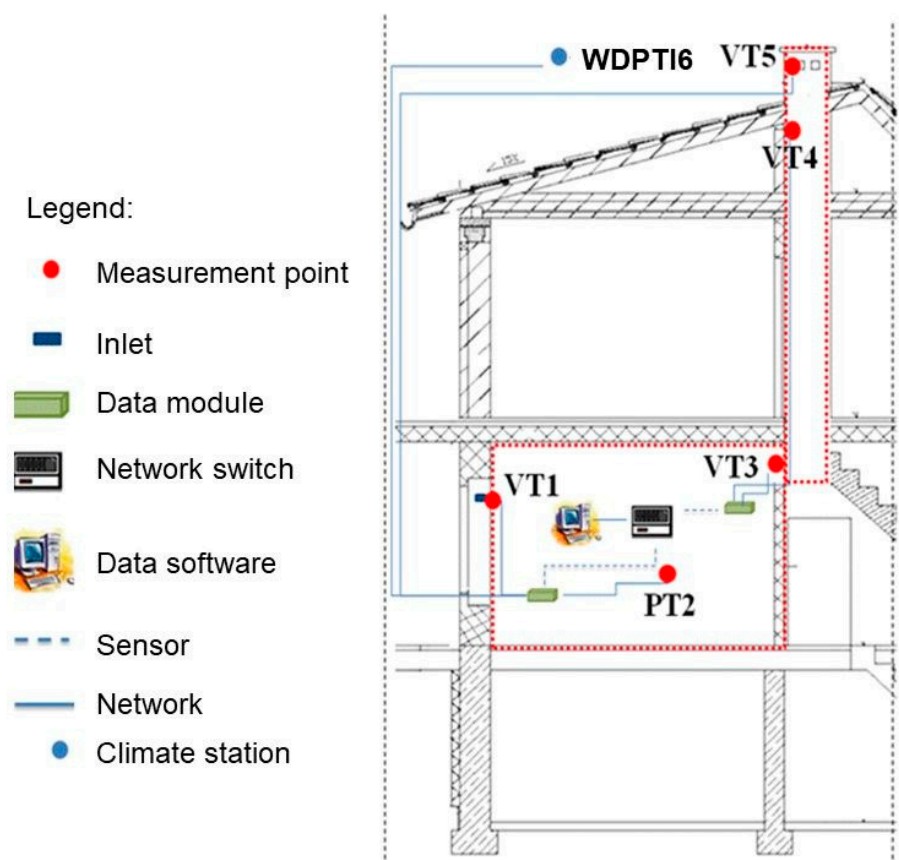

**Figure 6.** The measurement system in the test building.

The test apartment was divided into rooms with lightweight internal walls. The door between the rooms was open. Air velocity and temperature were registered in the vent inlet (V1 and T1) and outlet (V3 and T3). In the indoor zone, indoor climate conditions were monitored and measured. Air temperature (T2) was measured in the kitchen. The barometric pressure was registered in the kitchen (P2). All sensors were located in the rooms at the height of 1.6 m from the floor level. Thus, proper information about realistic climate data of the local climate conditions is important [14]. A Local Climate Station was built (WDPTI6);the climate station measured wind velocity W [m/s], wind direction WD [°], pressure P [Pa], relative humidity H [%] and temperature of the ambient air $T_{amb}$ [°C] as well as solar radiation $I_{sol}$ [W/m$^2$].

To perform the measurements, components of the LAB-EL products line [13] were used. At the climate station, the omnidirectional wind anemometer LB-747 with the measurement range of 0.5 m/s to 90 m/s and accuracy of 2% was used. The omnidirectional wind anemometer LB-747 of the climate station was located on the roof of the building at the height of 9.69 m above the ground level and 2.05 m above the roof ridge. The barometric pressure was measured with LB-716, which worked in the absolute pressure mode within the range of 2–200 kPa and accuracy of $\pm$0.5 kPa. Temperature and relative humidity were measured with the thermo-hygrometer LB-710R. Its temperature range was $-40$–85 °C with the accuracy of $\pm$0.1 °C and its relative humidity range was 99% with the accuracy of $\pm$2.0%. In the study, the visible radiation sensor type LB-901 (LITE2—Kipp&Zonen - The Netherlands) was used, with the measurement range of 0 W/m$^2$ to 2000 W/m$^2$ and an accuracy of 5%. In the indoor zones, barometric pressure was measured with LB-750, which worked in the absolute pressure mode within the range of 0.1–120 kPa and accuracy of $\pm$0.1 kPa. The CO2concentration was measured with LB-854 sensors. They measured the range of the CO2concentration from 0–10,000 ppm (accuracy $\pm$5%) and of the temperature from $-15$–100 °C (accuracy $\pm$0.2 °C). Air velocity and temperature were registered in the vent inlet and outlet. Due to complex airflow characteristics in the vent inlet (long rectangular shape) and potential flow direction changes, the air velocity measurement was difficult. In this measurement campaign, the thermo-anemometers LB-801A and LB-801C were installed in a double-sensor configuration. Both thermo-anemometers were based on a hot-wire concept. The first thermo-anemometer LB-801A worked in an analog mode and measured low velocity. The sensor could measure the indoor convection air velocity between 0.05–10 m/s with the accuracy of $\pm$0.05 m/s. The second thermo-anemometer LB-801C worked in a digital mode and was used to detect airflow direction. The airflow direction detection function was designed by a modification of the LB-801C sensor. The sensor was modified by closing its airflow channel from one side and re-calibrating its characteristics for two opposite flow directions. The new characteristics were recorded in its EEPROM memory. Both sensors were installed as an integrated measurement point. The measurement data management software LBX 2012 registered the flow direction and air velocity. All sensors were calibrated by the LAB-EL Laboratory in Poland before their installation and commissioning. The sensors were confirmed by means of suitable calibration certificates. Before the start of the measurements, there were several days of testing. The sensors used in the experiment are presented in Table 3.

**Table 3.** The sensors used during in experiment.

| LB-801 | LB-716 | LB-710R | LB-747 | LB-750 | LB-901 |
|---|---|---|---|---|---|

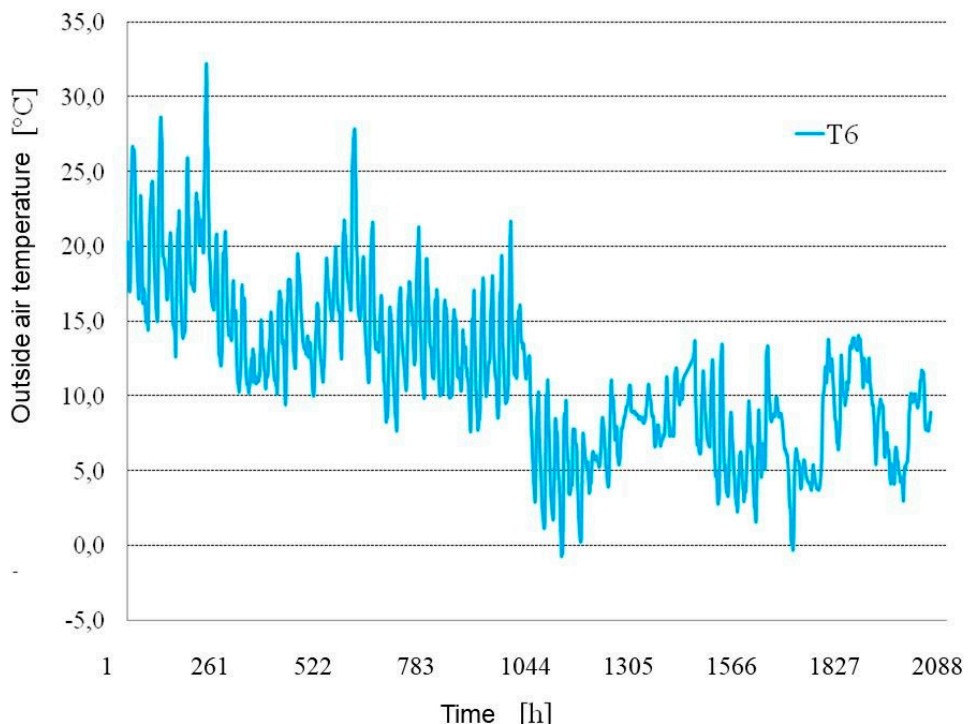

## 4. Climate Conditions

The research was conducted from 24 August to 19 November. To present the results more clearly, the measurements carried out with the frequency of 60 s were converted into the average value expressed in hours. Therefore, the graphs show the measurement results from 1 to 2088 h. The measurement of the outside air temperature showed that during the tests, the maximum outside temperature reached the value of 32.8 °C, while the minimum temperature was −0.4 °C (Figure 7). The mean value of the outside air temperature during the tests was 11.4 °C.

**Figure 7.** Outside air temperature.

Measurement of the wind velocity around the test building showed that weak winds prevailed during the tests, as the average wind velocity did not exceed 0.8 m/s. The maximum velocity during the measurements was 6.4 m/s and the minimum velocity was 0 m/s (Figure 8a). In the area of the test building, the dominant wind direction was north N (58%). A detailed wind rose with an average percentage of wind direction for the measured value is shown in (Figure 8b).

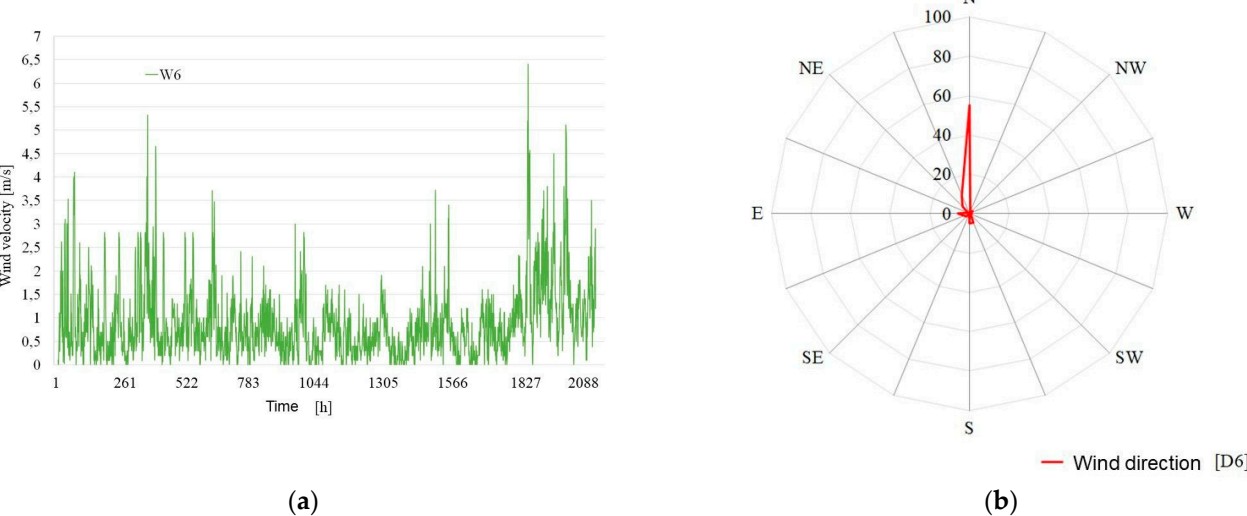

(**a**)

(**b**)

**Figure 8.** Information about wind: (**a**) wind velocity, (**b**) wind direction.

As the research was carried out from August to November, most of the measured radiation intensity during the day exceeded the value of 100 [W/m$^2$]. The measured intensity of solar radiation presented in (Figure 9) refers to the value of total solar radiation on the horizontal surface. Based on the measured values of solar radiation intensity from the NR LITE2 m (Kipp&Zonen), total solar radiation intensity Ic was obtained [15,16].

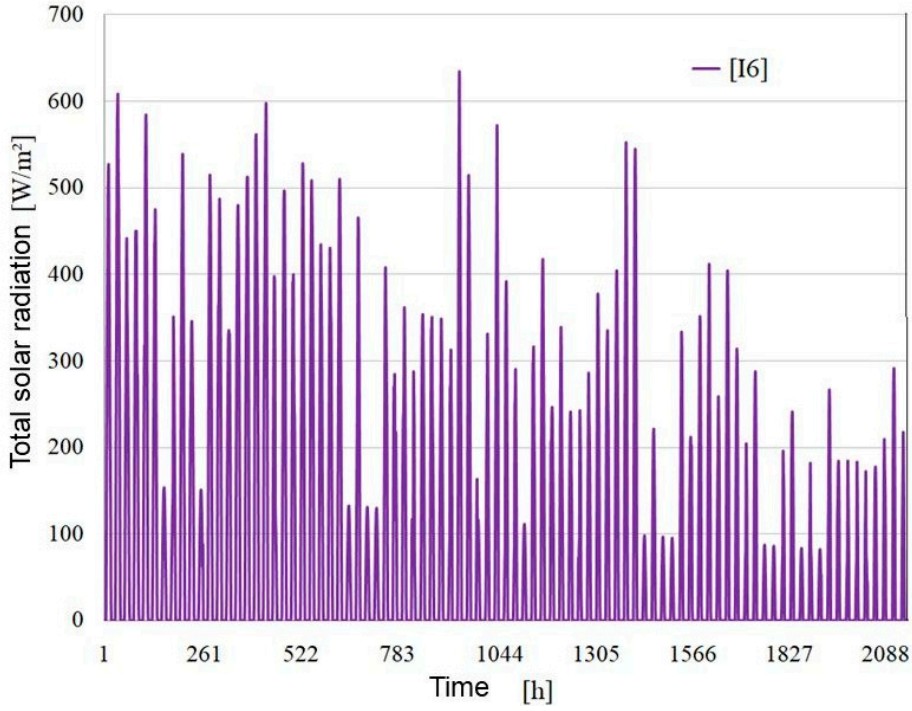

**Figure 9.** Total solar radiation during the experiment.

There is uncontrolled and unintentional airflow in buildings due to wind and density differences. This phenomenon is called air infiltration. Infiltration occurs in window gaps and microcracks in the building envelope. The airflow through windows, openings and gaps in the building envelope is the result of the sum of the pressure drops due to the wind and the temperature difference at the point of flow. In an isolated ventilation system, based on experimental tests, the value of uncontrolled airflow caused by infiltration was checked. To describe the infiltration, we need to know air density in the building test. So, during the experimental tests, the ventilation air density in the analyzed room was checked (Figure 10).

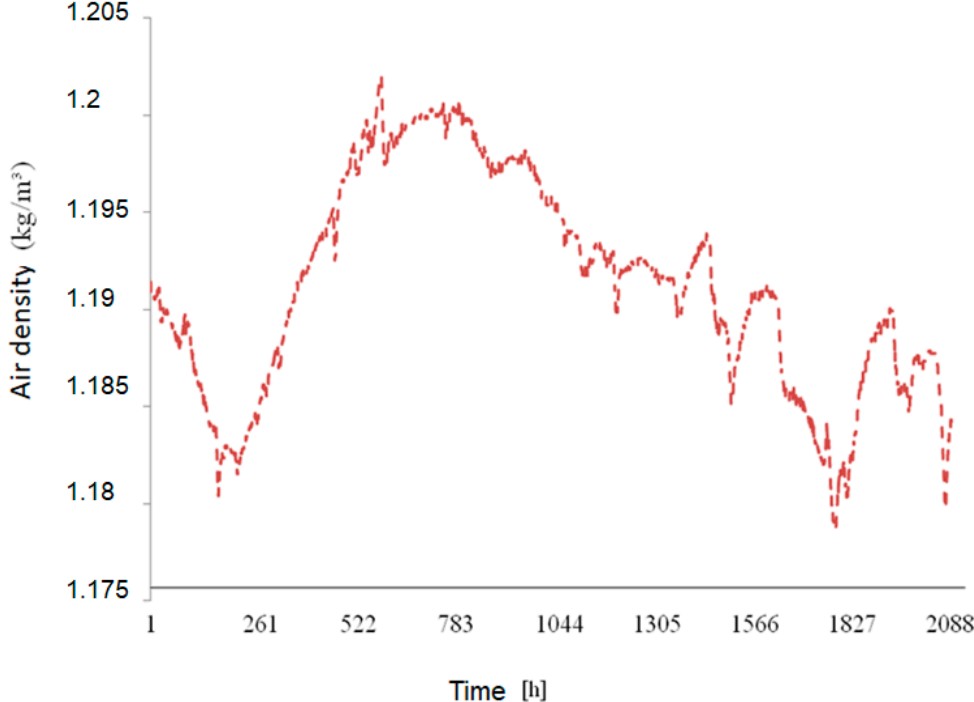

**Figure 10.** Air density in the analyzed building.

The average value of the ventilation air density in the analyzed room was 1.191 [kg/m$^3$]. Knowing the measurement of air velocity at the inlet to the room and at the outlet from the ventilation duct, we can determine the amount of infiltration air based on the mass balance. The general equations governing the mass conservation in each zone can be written as follows:

$$\frac{d\rho_i V}{dt} = \rho_{inlet}\left(A_{eff\ inlet}\ U_{inlet}\right) + \rho_{outlet}\left(A_{eff\ outlet}\ U_{outlet}\right) + \rho_{leak}q_{leak} = 0 \qquad (2)$$

where $\frac{d\rho_i}{dt}$ is the air density change of the indoor air during the time step $dt$, $\rho_{inlet}$ is the air density in the inlet gap, $A_{eff\ inlet}\ U_{inlet}$ is the volumetric flow rate in the inlet, $\rho_{outlet}$ is the air density of indoor airflow in the outlet, $A_{eff\ outlet}\ U_{outlet}$ is the volumetric flow rate in the outlet, $\rho_{leak}$ is the air density of leakage and $q_{leak}$ is the flow rate of leakage. In the study, it is assumed that $V$ is constant and equal to the total volume of the test apartment which is 40.95 m$^3$. In the opposite of the stack ventilation, the infiltration rate refers to the air change rate through unintentional leakage areas of the building envelope when doors and windows are closed. As the types of cracks in a specific building are not known, air leakage is difficult to estimate. In order to estimate the leakage flow rate, measured volume was used to detect it: the time-average flow rate in the air outlet ($A_{eff\ outlet}\ U_{outlet}$) and the time-average flow rate in the air inlet ($A_{eff\ inlet}\ U_{inlet}$) (Figure 11).

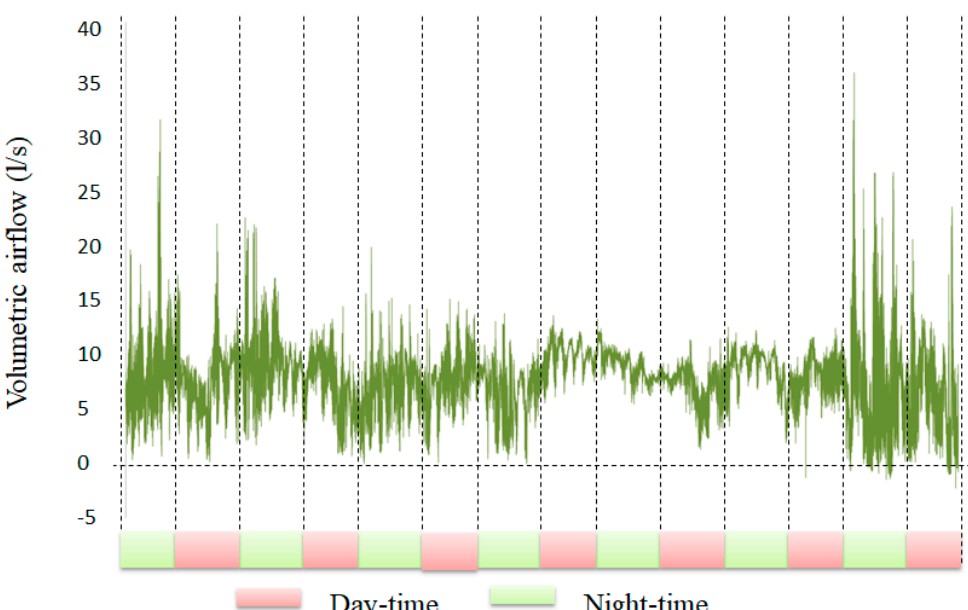

**Figure 11.** Time-variation of leakage flow rate in test apartment.

The analysis shows that the maximum value of infiltration is 0.035 ($m^3/s$), which corresponds to the maximum multiplication of air changes ACH from infiltration equal to 2.05 [1/h]. The average value of infiltration is 0.008 ($m^3/s$). However, the mean ACH value since infiltration did not exceed 0.44 [1/h]. Research by Shi et al. [10] has shown that the amount of infiltration is strongly dependent on the age of the building. In their research, the authors analyzed buildings built before 1990 and concluded that the amount of ACH caused by infiltration may range from 0.06 to 0.82 [1/h]. The test building where the research was conducted, which was built before 1990 and underwent partial modernization in 2012, has a low ACH value from infiltration. The obtained ACH value from infiltration is related to the isolation of the ventilation system in the form of a room + duct from the entire building. The low ACH value also proves good isolation of this system. Since in this case, the infiltration ACH value will not have a large influence on the ventilation ACH, this value is omitted later in the paper.

During the experimental tests, the direction of the ventilation airflow was measured in an isolated system. The test consisted in placing two thermoanemometers in a ventilation grill. During the measurements, one of the sensors was shielded from the side of the ventilation grill in order to check whether the ventilation sequence was reversed during the measurements. In Figure 12, which shows the course of the variability of the air temperature inside the room, the temperature measured at the vent with two sensors was very similar. A difference of 1 °C was observed, which proves that there is no reverse draft of ventilation air in the system.

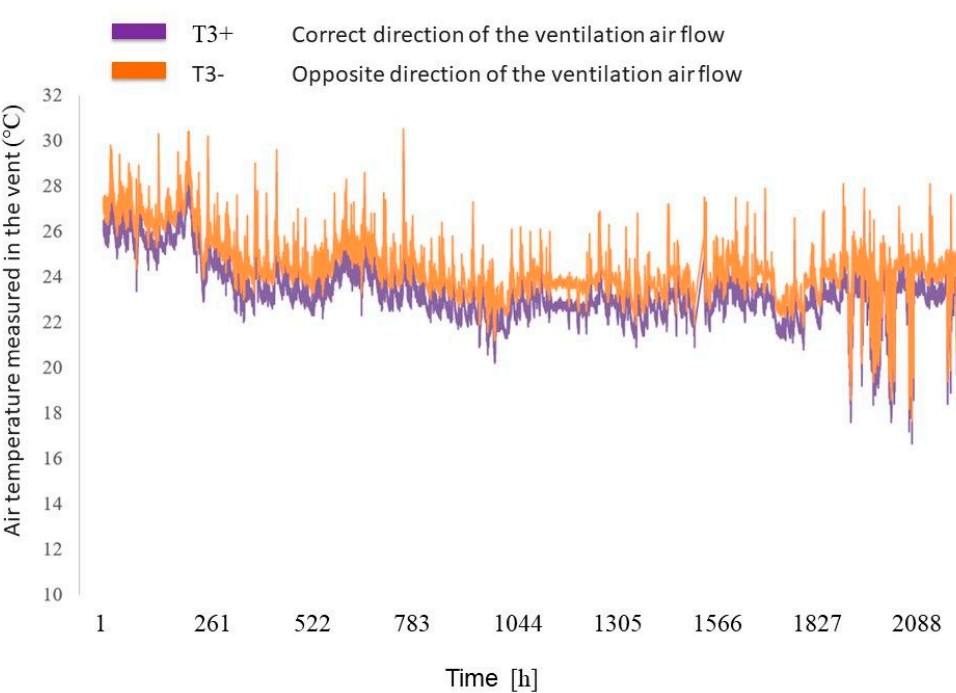

**Figure 12.** Air temperature measured in the vent.

## 5. Measurements of Gravity Ventilation Efficiency (ACH)

The experimental research was aimed at determining the efficiency of gravity ventilation (ACH) in transient conditions, i.e., in conditions where the temperature of the outside air is too high to cause air movement in the system by natural convection, and when the wind velocity is too weak to convection forced air movement in the ventilation system.

Based on the conducted experimental studies, the efficiency of gravity ventilation (ACH) for TYPE I was estimated for two different conditions of the external climate. The ACH efficiency was measured for the time interval in which the dominant air exchange mechanism in the ventilation system was wind—in the first approach, and the intensity of sunlight—in the second approach. The measured ACH performance allowed us to compare the results with type II and type III.

In the measurement of ACH when the dominant mechanism was wind, for the purposes of the analysis, the following assumptions were made:According to [6], stack ventilation may not work properly if the outside air temperature exceeds 12 °C.

Based on previous research [17], it was noticed that when the wind velocity remained within the range of 0–3 m/s, that is, when we talk about wind silence or so-called light wind, it does not contribute to the movement of ventilation air in the rooms. The analysis of the efficiency of gravity ventilation was extended by the wind velocity range from 2 to 5 m/s. The intensity of solar radiation did not exceed the value of 50 W/m$^2$.

The evaluation of the operation of the stack ventilation system was made based on continuous measurements. Measurements for the traditional system were carried out from August to September. The sensor recording the airflow in the duct was placed at the duct outlet. To determine ACH, the ventilation air velocity at the outlet from the duct was assumed. The average value of the ventilation air velocity at the outlet from the V5 was 0.59 m/s. The maximum value of the air velocity was 1.79 m/s and it corresponded to the maximum value of ACH = 2.03 1/h. On the other hand, the minimum value of the air velocity was V = 0.05 m/s and corresponded to the minimum value of ACH = 0.09 1/h (Figure 13). The average value of the gravity ventilation efficiency for a traditional chimney is 0.67 1/h.

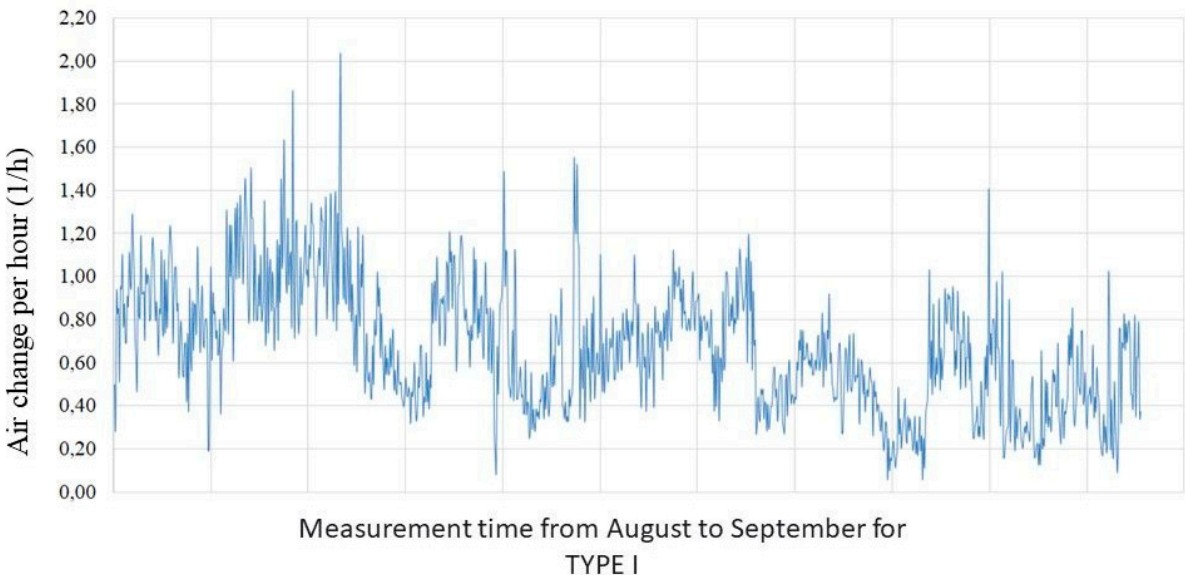

**Figure 13.** Gravity ventilation efficiency—chimney TYPE I. ACH measurement when the dominant mechanism forcing the ventilation air movement was wind.

For the ACH measurement when solar radiation intensity was the dominant air exchange mechanism in the ventilation system, the following assumptions were made:

- Outside air temperature above 12 °C.
- The wind velocity range from 0 to 2 m/s was adopted for the analysis.
- The intensity of solar radiation is above 100 W/m².

The average value of the ventilation air velocity at the outlet from the V5 was 0.48 m/s. The maximum value of the air velocity was V = 1.47 m/s and it corresponded to the maximum value of ACH = 1.67 1/h. On the other hand, the minimum value of air velocity was V = 0.02 m/s and corresponded to the minimum value of ACH = 0.02 1/h (Figure 14). The average value of the gravity ventilation efficiency for a traditional chimney was 0.61 1/h.

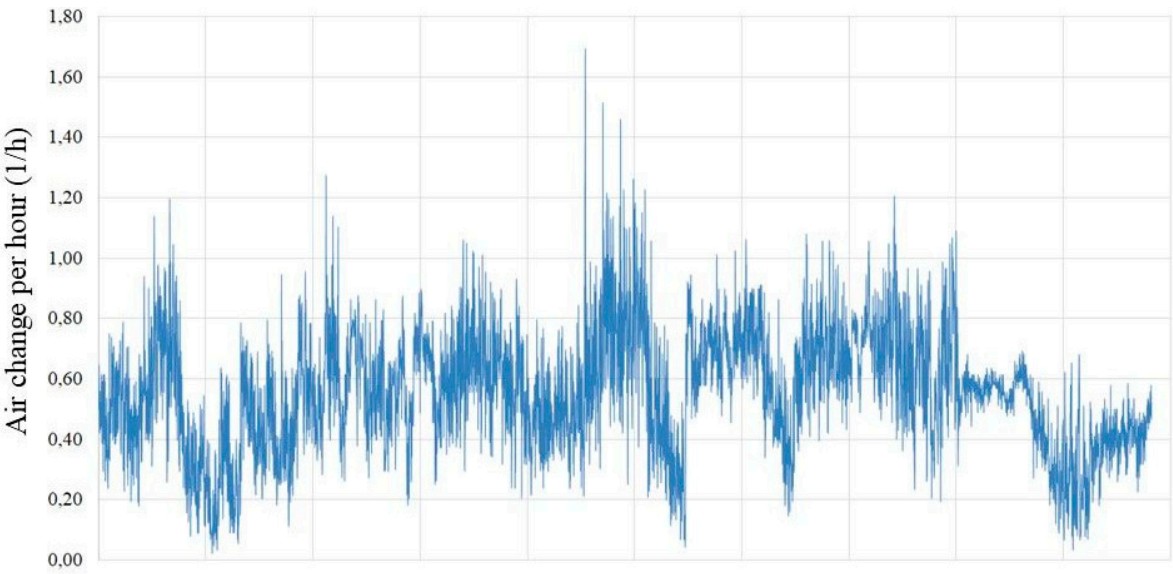

**Figure 14.** Gravity ventilationefficiency—chimney TYPE I. ACH measurement when the dominant mechanism was the solar radiation.

### 5.1. ACH Measured for TYPE II—Solar Chimney

For the purposes of the analysis, the following assumptions were made:

- Outside air temperature above 12 °C.
- The wind velocity range from 0 to 2 m/s was adopted for the analysis.
- The intensity of solar radiation is above 100 W/m$^2$.

The average value of the ventilation air velocity at the outlet from the V5 duct was 0.56 m/s. The maximum value of the air velocity was V = 1.79 m/s and it corresponded to the maximum value of ACH = 2.03 1/h. On the other hand, the minimum value of air velocity was V = 0.01 m/s and corresponded to the minimum value of ACH = 0.01 1/h (Figure 15). The average value of the gravitational ventilation capacity for the solar chimney is 0.71 1/h.

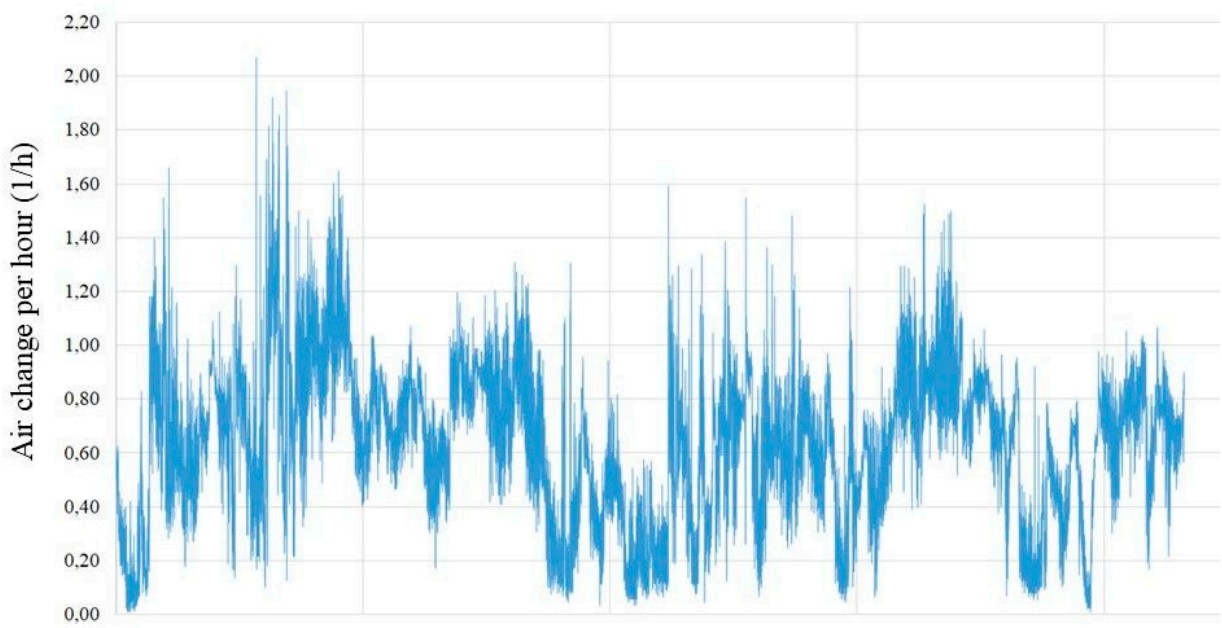

**Figure 15.** Gravity ventilation efficiency—solar chimney TYPE II.

### 5.2. ACH Measured for TYPE III—Chimney Cap

Stack ventilation works best when the temperature outside is much lower than in the building [18], if it is close to the temperatureinside, the draft force in the ventilation ducts becomes too low to effectively remove air from the rooms [19]. The spring-autumn transition period is the worst in terms of stack ventilation efficiency [17]. To improve the technical efficiency, understood as efficiency, of stack ventilation, among others, chimney caps are used. They are designed to improve the chimney effect. The research [20] shows that in the period of negative temperatures, the efficiency of gravitational ventilation is at a satisfactory level. However, based on research [17], it is known that during the occurrence of wind, gravitational ventilation and more precisely the rate of air changes, is three times higher than in the case of windless weather. For the purposes of the analysis, the following assumptions were made:

- Outside air temperature $\geq$12 °C.
- The wind velocity range from 2 to 5 m/s was adopted for the analysis.
- The intensity of solar radiation $\leq$50 W/m$^2$.

To improve gravity ventilation in the building, a rotary type of chimney cap was installed. The evaluation of the operation of the stack ventilation system was made based

on continuous measurements. Measurements for the system with a chimney cap were carried out from October to November. The sensor recording the airflow in the ventilation system was placed at the outlet of the vent. The average value of the ventilation air velocity at the outlet from the V5 vent was 0.75 m/s. The maximum value of the air velocity was V = 2.15 m/s and it corresponded to the maximum value of ACH = 2.44 1/h. However, the minimum value of the air velocity was V = 0.1 m/s and corresponded to the minimum value of ACH = 0.11 1/h (Figure 16). The average value of the gravitational ventilation efficiency for a chimney with a chimney is 0.85 1/h.

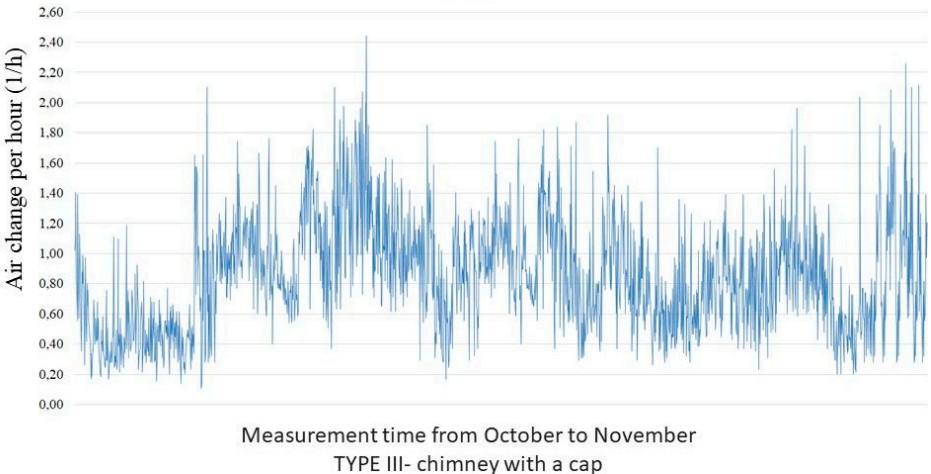

**Figure 16.** Gravity ventilation efficiency—chimney with a cap TYPE III.

The research shows that the rotary-type chimney increases the efficiency of gravity ventilation for the outside temperature above 12 °C and the wind velocity of 2–5 m/s compared to the chimney without the root by about 35% on average. Based on the conducted research, it was noticed that for the maximum values, the chimney cap improves the draft in the ventilation duct by 0.40 1/h (Figure 17). However, in the case of a solar chimney for an outside temperature above 12 °C, a wind velocity of 0–2 m/s and a solar radiation intensity above 100 W/m² , we notice an average improvement in ventilation efficiency by 14% than in the case of a traditional chimney (for the same external climate conditions). For the maximum values, the solar chimney improves the draft in the ventilation duct by 0.36 1/h (Figure 17).

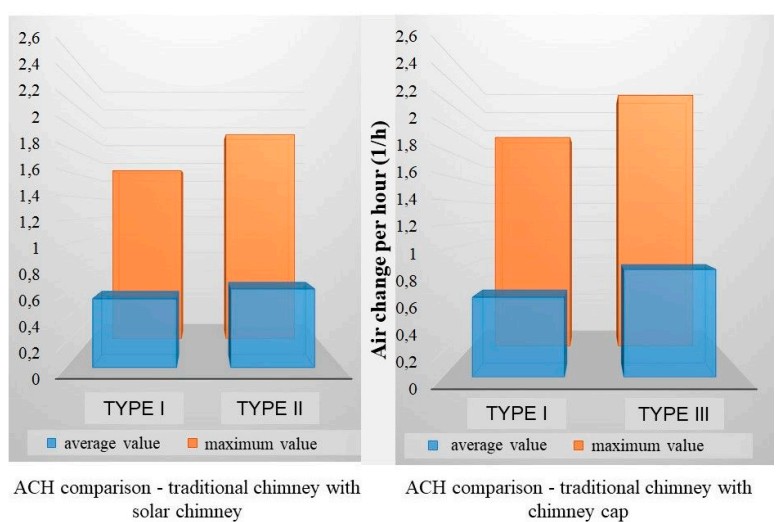

**Figure 17.** Summary of ACH results average value and maximum value for the three types of chimneys.

To summarize, it can be stated that the scope of the research was sufficient to notice the influence of different types of chimneys on ACH. Due to the popularity of the solutions and the effect obtained, it was found that the chimney cap is the optimal solution to improve gravity ventilation in this type of buildings.

Additionally, ACH results for windy and no wind periods were compared. Based on the results presented in Figure 18, we can observe a clear influence of the chimney cap on the air exchange rate in the room. In the case of a windless period, all three types of chimneys achieve an approximate ACH value (Figure 18).

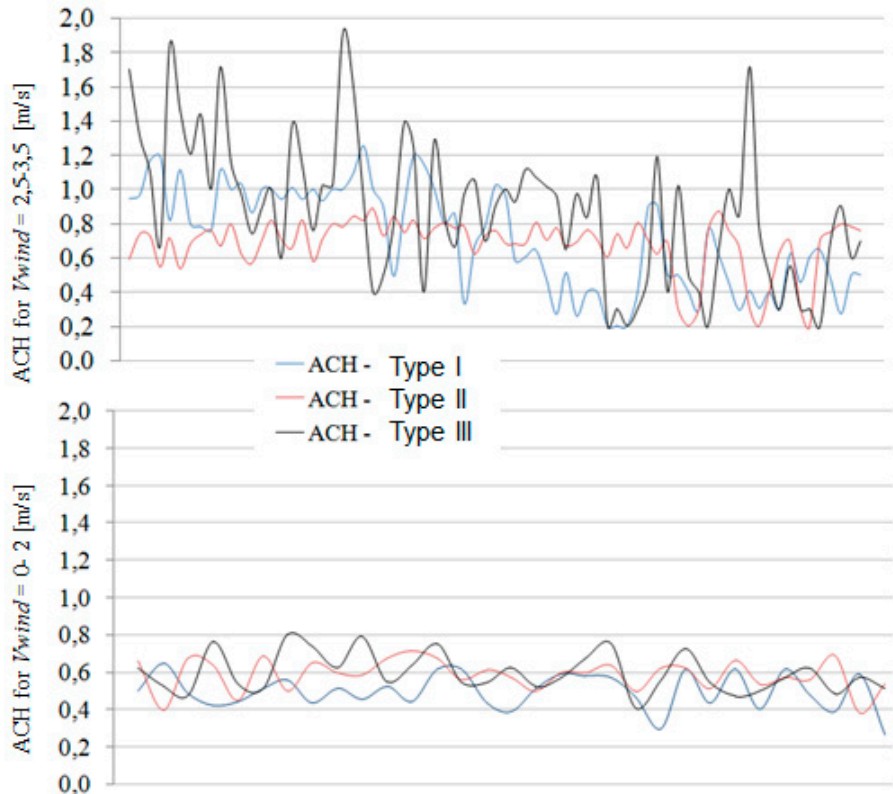

**Figure 18.** Summary of ACH results during windy and no windy periods.

The ACH results for sunny and cloudy periods were also compared. Based on the results presented in Figure 19, we can see the dominant influence of the solar chimney on the air exchange in the room. In the case of the cloudy period, the ACH value for the solar chimney reduces by approx. 0.4 1/h, approaching the times of air exchange obtained by the other two chimneys (TYPE I and TYPE III) (Figure 19).

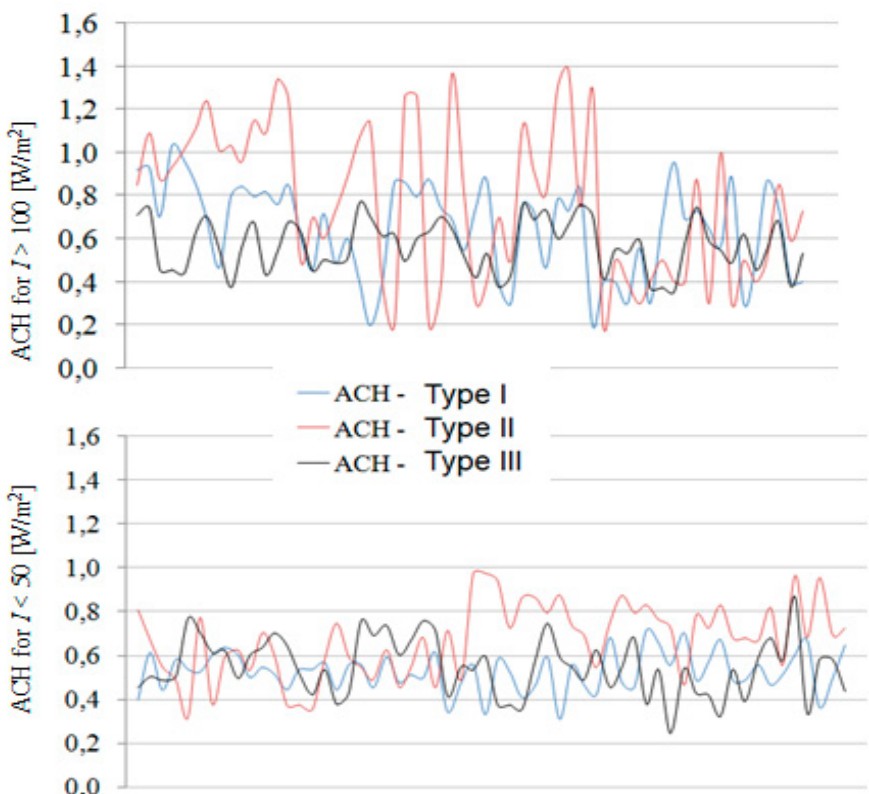

**Figure 19.** Summary of ACH results during sunny and cloudy periods.

In the case of a solar chimney, additional results at different solar radiation values were presented.Based on Figure 20 we can observe a direct influence of solar radiation on ventilation efficiency (ACH). As the radiation increases, usually the ventilation efficiency (ACH) enhances. After the research cycle, it can be concluded that solar radiation undoubtedly improves the efficiency of ventilation with a solar chimney, but it is difficult forthis research to precisely determine the degree of influence.

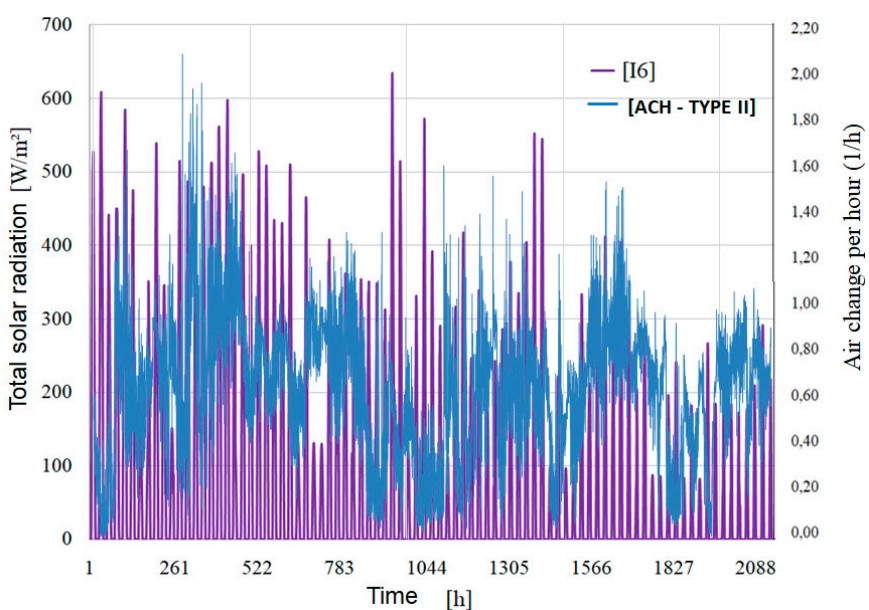

**Figure 20.** Experimental comparison of solar radiation with ACH for TYPE II.

## 6. Assessment of Methods for Improving the Gravity Ventilation System

A frequent problem in gravity ventilation isdisturbances in the thrust in ventilation ducts. This affects the lack of adequate air exchange in the building, which results in moisture and poor air quality. This has a negative impact on the well-being and health of people. Due to the phenomenon of weakening the chimney draft in buildings, the article proposes technical solutions to strengthen the draft. The proposed solutions to strengthen the chimney draft are:

- Solar chimneys
- Chimney caps

The technical and economic analysis of the solutions used to improve the chimney draft is presented below. In the technical part, the method of production and availability were analyzed, while in the economic part, the costs of manufacturing and assembly as well as operating the building with a chimney pot were checked.

The following factors were used to assess the effectiveness of the solutions used to improve the chimney draft in the ventilation system:

(a) Technology availability;
(b) The cost of materials and labor;
(c) Execution time;
(d) Aesthetic qualities.

Based on the above factors, the methods used to improve the chimney draft were assessed. A five-point scale was used for the evaluation, where: 1—means the lowest grade, and 5—means the best grade.

### 6.1. Solar Chimney

In this case, the solar chimney consists of two basic installations: an air solar collector and a ventilation chimney made of ceramic bricks. The effectiveness of the applied solution is presented in tables Table 4.

**Table 4.** Self-assessment of the applied solution to strengthen the chimney draft.

| Factor | Comment | Assessment |
|---|---|---|
| Availability of technology | Typical, readily available building materials were used for the construction of the solar chimney. For the construction of a transparent partition, colorless plexiglass, 4 mm thick, cut to size was used. The joints are made of 3 × 3 cm wooden slats. | 4 |
| Cost of materials and labor | The purchased material cost PLN 320, while the cost of assembly and then assembly work was calculated at PLN 250, which gives a total amount of PLN 570. | 3 |
| Time of implementation | The time to assemble the structure took two working days + one day for assembly. | 3 |
| Aesthetic values | In this case, the structure of the chimney lining was clearly visible from the street side. | 3 |
| Total points: | | 13 |

### 6.2. Chimney Cap

There isa passive or active chimney cap with an additional electric drive. Passive chimney cowls (e.g., rotating) are constructed in such a way that, at a low wind speed (>1 m/s), they generate negative pressure in the chimney outlet while maintaining the correct airflow direction. The effectiveness of the applied solution is presented in tabular form. Table 5.

**Table 5.** Assessment of the applied solution to strengthen the chimney draft.

| Factor | Comment | Assessment |
|---|---|---|
| Availability of technology | The chimney cowl is a generally available product on the market. Due to the quality, price and purpose, there are several dozen types of chimney cowls. Without leaving your home, you can choose the right type of attachment by talking to a technical advisor. | 5 |
| Cost of materials and labor | The cost of the attachment depends on the type: fixed attachments cost 60–250 PLN, self-adjusting attachments 170–450 PLN, and rotary ones 180–500 PLN. In this case, a rotating cowl was selected, the cost of which, including assembly, was PLN 480. | 5 |
| Time of implementation | The element was delivered and installed within one day. | 5 |
| Aesthetic values | In this case, from the side of the street, the attachment on the chimney was also clearly visible. | 3 |
| | Total points: | 18 |

Looking at single-family buildings and collective housing, it was noticed that in the temperate climate zone typical of the Pomeranian Voivodeship, the most popular methods of improving gravity ventilation include the chimney cap. Based on the observations, it was found that the most popular chimney cap, used mainly in multi-family housing, is the fixed type (Figure 21a). This device is the simplest in terms of construction and the cheapest type of attachment. Among single-family housing, it happens often that residents decide to use a rotary type. Based on the observations, it is difficult to determine whether the attachment has an additional motor (Figure 21b).

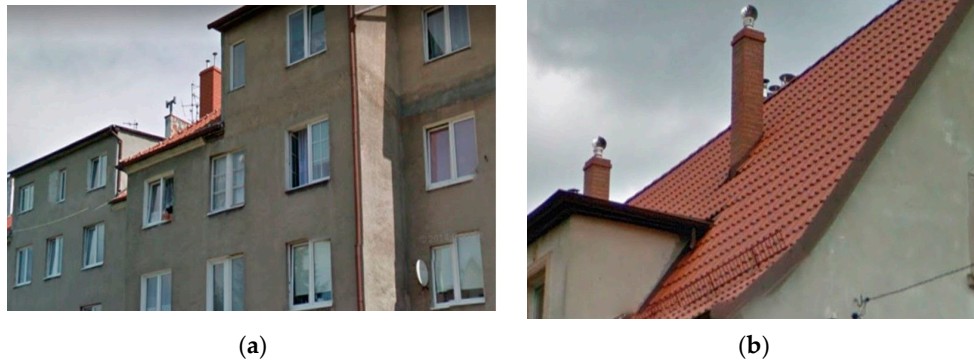

(**a**)　　　　　　　　　　　　　　　　　　(**b**)

**Figure 21.** Multi-family housing in Gdańsk: (**a**) traditional chimney cap and (**b**) rotary type chimney cap.

In the paper compares the effect of a chimney cap on the heating properties of a building. Based on the experimental tests, the efficiency of the device was checked by comparing the consumption of thermal energy in the building (Figure 22). The results were obtained by measurements.

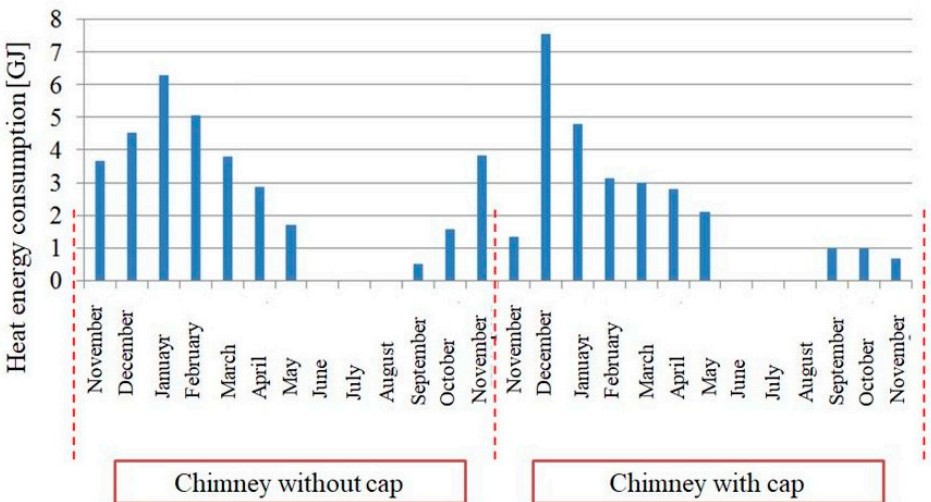

**Figure 22.** Comparison of thermal energy consumption for heating a building.

Based on the diagram (Figure 22), the effect of the chimney cap on the consumption of thermal energy for heating was compared. A total of 24 months were adopted for the analysis, where the first 12 months are the result without the chimney cap, and the next 12 months are the measurement with the chimney cap installed. The results showed that the annual heat energy consumption in the first case (without the cap) was 33.74 [GJ], while the annual heat energy consumption with the chimney cap decreased to 27.21 [GJ]. The conducted experiments show that the chimney cowl gave savings of about 6.53 [GJ]. This is because the chimney cap constantly maintains the proper draft in the chimney, reducing it when the wind is too strong, and raising it when the wind is weak, so that the operating conditions of the installation are close to the parameters defined by the manufacturer. In addition, the chimney remains dry along its entire length. The effect of the chimney cap on the stabilization of the air flow conditions can be seen in (Figure 22), where, in the windy months (March, October, November), the measurements with the chimney cap recorded lower heat energy consumption.

## 7. Conclusions

Technical solutions strengthening the chimney draft consisting in the assembly of a rotary chimney cap (TYPE III) and the construction of the so-called solar chimney (TYPE II) werecompared with the experimental results carried out on a traditional chimney (TYPE I). The research shows that the rotary-type chimney increases the efficiency of gravity ventilation (for the assumptions $T_e > 12$ °C and $V_{wind} = 2–5$ m/s) compared to the chimney without the cap by an average of about 27%. On the other hand, for the maximum values, the chimney cowl improved the draft in the ventilation duct even by approx. ACH = 0.40 [1/h]. In the case of the solar chimney (for the assumptions of $T_e > 12$ °C, $V_{\_wind} = 0–2$ m/s and $I > 100$ W/m$^2$), the ACH improvement by 16% is noticeable compared to the traditional chimney (for the same external climate conditions). However, in the case of maximum values, the solar chimney improves the draft in the ventilation duct by about ACH = 0.36 [1/h]. The conducted research has shown that for the average values obtained during unfavorable external climate conditions, the chimney cap works best, the ACH value of which does not fall below 0.85 [1/h]. A necessary condition for the operation of the chimney cap is a slight wind blowing at a speed of not less than 2 m/s.

Assuming that in the Polish (especially northern) climate zone, windy days are the most common days in the calendar year, it can be concluded that the chimney cap will be more effective throughout the year than the solar chimney.Guidelines for the design of gravity ventilation include:

-  Building location: on a hill, in a valley, etc.

- Shading/obstacles near the building.
- Considering air infiltration in the air balance of the apartment.
- Considering the opening of windows as an activity ensuring an increase in the intensity of gravity ventilation.
- Consideration of maintenance issues during designing gravity ventilation.

Based on the assumed factors, two solutions to improve the ventilation air flow in buildings were assessed. The biggest differences in scores were noted for the cost and time of construction in favor of the chimney cap.

Technically:

- The chimney cowl is easier to install.
- The availability of technical solutions is greater in the case of the root.

Economic:

- Due to the large number of companies involved in the production of chimney caps, the chimney cap is a cheaper solution to improve the chimney draft.

The obtained results showed that the chimney cowl obtains the point advantage in terms of engaging the smallest possible resources, especially time and money.

Based on the observations, no methods other than the chimney cap improving the efficiency of gravity ventilation were noticed. A chimney cap is a generally available product with its assembly, while solar chimneys have not been commercialized, which may affect the choice of the method.

**Author Contributions:** Conceptualization, R.A.-J., K.P. and M.N.; methodology, R.A.-J.; formal analysis, R.A.-J., K.P. and M.N.; investigation, R.A.-J., K.P. and M.N.; writing—original draft preparation, R.A.-J., K.P. and M.N.; supervision, K.P. and M.N. All authors have read and agreed to the published version of the manuscript.

**Funding:** This research received no external funding.

**Institutional Review Board Statement:** Not applicable.

**Informed Consent Statement:** Not applicable.

**Data Availability Statement:** Data are contained within the article.

**Conflicts of Interest:** The authors declare no conflict of interest.

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
