# Peer review of "Improvement of the Chimney Effect in Stack Ventilation"

_applsci, doi:10.3390/app11199185_

Round 1
Reviewer 1 Report
Paper presents different options of the ventilation in the building. Although the topic is interested, this is more "technical" that "scientific" paper.
Some improvements are needed:
- pictures on Figure 3 are "not clear";
- it is not clear why equations for solar radiation ( 5 - 9 ) are used since the research seems to be experimental);
- airflow (figure 14): better in l/s
- for "fair" comparison measurements at "equal" conditions needs to be made (which is not so hard to achieve);
- conclusion might give some suggestions for ventilation standards;
- A lot of references are in Polish (hard to understand for non polish readers);
Author Response
Dear Reviewer,
thanks to you for your good comments. All of your coments were included and hope that the correction will meet with approval. Please see the attachment.

Reviewer 2 Report
Please see the attachement.

Author Response
Dear Reviewer,
thanks to you for your good comments. According to your comments, we have shortened our discussion. We appreciate for Reviewer' warm work earnestly and make it more conciseand hope that the correction will meet with approval. Please see the attachment.

Round 2
Reviewer 2 Report
When answering reviewers' comments please add a word document that includes a description of the made changes or responses to the comments why no changes have been made. At the moment only the word version of the paper has been added. This causes extra work for the reviewing process. I see that some changes have been made but can not determine if everything is corrected.
